**Interpretation of Multi-scale Permeability Data through an Information Theory Perspective**

**Aronne Dell'Oca, Alberto Guadagnini, and Monica Riva**

Department of Civil and Environmental Engineering, Politecnico di Milano, 20133, Milano, Italy;

Corresponding author: Aronne Dell'Oca (aronne.delloca@polimi.it)

**Key Points**

- Information Theory allows characterizing information content of permeability data related to differing measurement scales.
- An increase of the measurement scale is associated with quantifiable loss of information about permeability.
- Redundant, unique and synergetic contributions of information are evaluated for triplets of permeability datasets, each taken at a given scale.

**Abstract**

We employ elements of Information Theory to quantify (*i*) the information content related to data collected at given measurement scales within the same porous medium domain, and (*ii*) the relationships among Information contents of datasets associated with differing scales. We focus on gas permeability data collected over a Berea Sandstone and a Topopah Spring Tuff blocks, considering four measurement scales. We quantify the way information is shared across these scales through (*i*) the Shannon entropy of the data associated with each support scale, (*ii*) mutual information shared between data taken at increasing support scales, and (*iii*) multivariate mutual information shared within triplets of datasets, each associated with a given scale. We also assess the level of uniqueness, redundancy and synergy (rendering, i.e., information partitioning) of information content that the data associated with the intermediate and largest scales provide with respect to the information embedded in the data collected at the smallest support scale in a triplet.

**Plain Language Summary**

Characterization of the permeability of a geophysical system, or part of it, is a key aspect in many environmental settings. Permeability of natural systems typically exhibits spatial variations and its spatially heterogeneous pattern is linked with the size of observation/measurement/support scale. As the latter becomes coarser, the system appearance is less heterogeneous. As such, sets of permeability data associated with differing support scales provide diverse amounts of information. In this contribution, we leverage on elements of Information Theory to quantify the information content of gas permeability datasets collected over a Berea Sandstone and a Topopah Spring Tuff blocks and associated with four measurement scales. We then characterize the nature of the information shared by the diverse datasets, in terms of redundant, unique and synergetic forms of information.

## 1. Introduction

Characterization of permeability of porous media plays a major role in a variety of hydrological settings. There are abundant studies documenting that permeability values and their associated statistics depend on a variety of scales, i.e., the measurement support (or data support), the sampling window (domain of investigation), the spatial correlation (degree of structural coherence) and the spatial resolution (rendering the degree of the descriptive detail associated with the characterization of a porous system) (see e.g., Brace 1984; Clauser, 1992; Neuman, 1994; Schad and Teutsch, 1994; Rovey and Cherkauer, 1995; Sanchez-Villa et al., 1996; Schulze-Makuch and Cherkauer, 1998; Schulze-Makuch et al., 1999; Tidwell and Wilson, 1999a, b, 2000; Vesselinov et al., 2001a, b; Winter and Tartakovsky, 2001; Hyun et al., 2002; Neuman and Di Federico, 2003; Maréchal et al., 2004; Illman, 2004; Cintoli et al., 2005; Riva et al., 2013; Guadagnini et al., 2013, 2018 and references therein). Among these scales, we focus here on the characteristic length associated with data collection (i.e., support scale).

In this context, experimental evidences at the laboratory scale (observation scale of the order 0.1-1.0 m) suggest that the mean value and the correlation length of the permeability field tend to increase with the size of the data support, the opposite trend being documented for the variance (e.g., Tidwell and Wilson, 1999a, 1b, 2000). Similar observations, albeit with some discrepancies, are also tied to investigations at larger scales (i.e., 10-1000 m) (Andersson et al., 1988; Guzman et al., 1994, 1996; Neumann, 1994; Schulze-Makuch and Cherkauer, 1998; Zlotnik et al., 2000; Bulter and Healey, 1998a,b). We consider here laboratory scale permeability datasets which are associated with various measurement scales.

The above mentioned documented pattern suggests that the spatial distribution of permeability
tends to be characterized by an increased degree of homogeneity (as evidenced by a decreased
variance and an increased spatial correlation) as the support/measurement scale increases. At the same
time, increasing the measurement scale somehow hampers the ability to detect locally low
permeability values, as reflected by the observed increased mean value of the data. As an example of
the kind of data we consider in this study to clearly document these features, Figure 1 depicts the
spatial distribution of the natural logarithm of (normalized) gas permeabilities, i.e., $Y_{r_i} = \ln(k_{r_i} / \bar{k}_{r_i})$
(where $k_{r_i}$ is gas permeability and $\bar{k}_{r_i}$ is the mean value of the data), collected across two faces of a
laboratory scale block of (*i*) a Berea Sandstone (Tidwell and Wilson, 1999a) and (*ii*) a Topopah Spring
Tuff (Tidwell and Wilson, 1999b) at four support scales $r_i$ (see Section 2 for a detailed description).
As a preliminary observation, one can note that increasing the measurement scale $r_i$ yields a
decreased level of descriptive detail of the heterogeneous spatial distribution of the system properties.
It is important to note that a reduced level of details in the description of the system properties (e.g.,
$Y_{r_i}$) could hinder reliability and accuracy of further predictions of system behavior (in terms of, e.g.,
flow and solute transport patterns). It is therefore relevant to quantify the amount of loss (or of
preservation) of the information about the system properties associated with a fine scale(s) of
reference as the data support increases.
Our study aims at providing an assessment and a firm quantification of these aspects upon
relying on Information Theory (IT) (e.g., Stone, 2015) and the multiscale collection of data described
above. We consider such a framework of analysis as it provides the elements to quantify (*i*) the
information content associated with a dataset collected at a given scale as well as (*ii*) the information
shared between pairs or triplets of data, each associated with a unique scale (while preserving the
design of the measurement device). In this context, IT represents a convenient theoretical framework
to properly assist the characterization of the way the information content is distributed across sets of
measurements, without being confined to a linear analysis (relying, e.g., on analyses of linear
correlation coefficients) or invoking some tailored assumption(s) about the nature of the
heterogeneity of permeability (e.g., the characterization of the datasets through a Gaussian model).
To the best of our knowledge, as compared to surface hydrology systems only a limited set of
works consider relying on IT concepts to analyze scenarios related to processes taking place in
subsurface porous media. Nevertheless, we note a great variety in the topics covered in these works,
reflecting the broad potential for applicability of IT concepts. These studies include, e.g., the works
of Woodbury and Ulrych (1993, 1996, 2000) who apply the principle of minimum relative entropy
to tackle uncertainty propagation and inverse modeling in a groundwater system. The principle of
maximum entropy is employed by Gotovac et al. (2010) to characterize the probability distribution
function of travel time of a solute migrating across a heterogeneous porous formation. Within the
same context, Kitanidis (1994) leverages on the definition of entropy and introduces the concept of
dilution index to quantify the dilution state of a solute cloud migrating within an aquifer. Mishra et
al. (2009) and Zeng et al. (2012) evaluate the mutual information shared between pairs of (uncertain)
model input(s) and output(s) of interest, and view this metric as a measure of global sensitivity.
Nowak and Guthke (2016) focus on sorption of metals onto soil and the identification of an optimal
experimental design procedure in the presence of multiple models to describe sorption. Boso and
Tartakovsky (2018) illustrate an IT approach to upscale/downscale equations of flow in synthetic
settings mimicking heterogeneous porous media. Relaying on IT metrics, Butera et al. (2018) assess
the relevance of non-linear effects for the characterization of the spatial dependence of flow and solute
transport related observables. Bianchi and Pedretti (2017, 2018) developed novel concepts, mutuated
by IT, for the characterization of heterogeneity within a porous system and its links to salient solute
transport features. Wellman and Regenaur-Lieb (2012) and Wellman (2013) leverage on IT concepts
to quantify uncertainty, and its reduction, about the spatial arrangement of geological units of a
subsurface formation. Recently, Mälicke et al. (2019) combine geostatistic and IT to analyze soil
moisture data (representative of a given measurement scale) to assess the persistence over time of the
spatial organization the soil moisture, under diverse hydrological regimes.
Here, we focus on the aforementioned datasets of Tidwell and Wilson (1999a, b) who conducted
extensive measurement campaigns collecting air permeability data across the faces of a Berea
Sandstone and a Topopah Spring Tuff blocks, considering four different support/measurement scales
(see Section 2 for details). While our study does not tackle directly issues associated with the way
one can upscale (flow or transport) attributes of porous media, we leverage on such a unique and truly
multiscale datasets to address research questions such as "How much information about the natural
logarithm of (normalized) gas permeabilities is lost as the support scale increases?" and "How
informative are data taken at a coarser support scale(s) with respect to those associated with a finer
support scale?" (see Section 3). In this sense, our study yields a unique perspective of the assessment
of the value of hydrogeological information collected at differing scales.

## 2. Dataset

We consider the datasets provided by Tidwell and Wilson (1999a, b), who rely on a
multisupport permeameter (MSP) to evaluate spatial distributions of air permeabilities across the
faces of a cubic block of Berea Sandstone (hereafter denoted as Berea) and Topopah Spring Tuff
(hereafter denoted as Topopah). Data are collected at uniform intervals with spacing $\Delta = 0.85$ cm
across a grid of $24 \times 24$ and $36 \times 36$ nodes along each face (of uniform side equal to 19.5 cm and
29.75 cm, to avoid boundary effects) of the Berea and the Topopah blocks, respectively. Four
measurement campaigns are conducted, each characterized by the use of a MSP with a tip-seal of
inner radius $r_i$ ($i = 1, 2, 3, 4$) = (0.15, 0.31, 0.63, 1.27) cm and outer radius $2r_i$ (interested readers can
find additional details about the MSP design and functioning in Tidwell and Wilson, 1997). While
the precise nature and size of the support/measurement scale associated with a MSP is still under
study for heterogeneous media (e.g., Goggin et al., 1988; Molz et al., 2003; Tartakovsky et al., 2000),
hereafter we denote data associated with a given support/measurement scale by referring these to the
associated value of $r_i$. The ensuing dataset is then composed by 3456 and 6480 data points for each
measurement scale, $r_i$, for the Berea and the Topopah block, respectively (we exclude data for one
of the faces of the Topopah block, due to some anomalies with respect to the other faces). We consider
here the quantity $Y_{r_i} = \ln(k_{r_i} / \bar{k}_{r_i})$, i.e., the natural logarithm of the air permeability normalized by the
mean value (i.e., $\bar{k}_{r_i}$) of the data of the corresponding sample.
The two types of rocks analyzed display distinct features. The Berea sample may be classified
as a very fine-grained, well-sorted quartz sandstone. Following Tidwell and Wilson (1999a), visual
inspection of the spatial distributions of $Y_{r_i}$ (see, e.g., Figure 1) shows that the Berea sample exhibits
a generally uniform spatial organization of permeabilities, devoid of particular features, with the
exception of a mild stratification, thus allowing to consider this sample as a fairly homogenous
system. Otherwise, the Topopah rock sample clearly exhibits a heterogenous structure whereas
pumice fragments ($\sim 23\%$ of the sample) are embedded in the surrounding matrix (see Figure 1). In

general, the pumice is characterized by higher permeability values than the surrounding matrix. As such, the Topopah sample can be considered as a fairly heterogenous system, with a tendency to display a bimodal distribution of permeability values (see also Section 4.2). In this sense, the two rock samples analyzed provide two clearly distinct scenarios for the analysis of the interplay of the information contained in datasets collected at diverse measurement scales.

We note that the IT elements described in Section 3 refer to discrete variables. While corresponding definitions are available also for continuous variables (i.e., summation(s) and probability mass function(s) are replaced by integral(s) and probability density function(s), respectively), these are characterized by a less intuitive and immediate interpretation (e.g., Entropy could be negative, infinite or could not be evaluated in case of probability density function(s) involving a Dirac's delta; see, e.g., Kaiser and Schreiber, 2002; Cover and Thomas, 2006). Moreover, in case the probability density functions of the analyzed continuous variables cannot be associated with an analytical expression, it is necessary to subject these variables to quantization and the IT metrics related to the continuous variables are estimated through their quantized counterparts (see Cover and Thomas, 2006). In general, the quality of these estimates increases (in a way which depends on the specific metric) with the level of quantization of the continuous variables (see, e.g., Kaiser and Schreiber, 2002). This leads us to treat $Y_{r_i}$ as a discrete variable, a modeling choice which is consistent with several previous studies (see, e.g., Ruddell and Kumar, 2009; Goodwell et al., 2017; Nearing et al., 2018 and references therein).

### 3. Methodology

### 3.1 Information Theory

Considering a discrete random variable, $X$, one can quantify the associated uncertainty through the Shannon entropy

$$H(X) = \sum_{i=1}^{N} p_i \log_2(p_i^{-1}) \tag{1}$$

where $N$ is the number of bins used to analyze the outcomes of $X$; and $p_i$ is the probability mass function and $\ln(p_i^{-1})$ is the (so-called) Information (or degree of surprise) associated with the $i$-th bin (see, e.g., Shannon, 1948). We employ base two logarithms in (1), thus leading to *bits* as unit of measure for entropy and for the IT metrics we describe in the following. While other choices (relying, e.g., on the natural logarithm) are admissible, the nature and meaning of the metrics we illustrate remain unaffected. The Shannon entropy can be interpreted as a measure of the uncertainty associated with $X$, i.e., $H(X)$ is largest and equal to $\log_2(N)$ in case $p_i$ is uniform across all bins (i.e., $p_i = 1/N$), while it is zero when outcomes of $X$ reside only within a single bin. Moreover, one can note that Shannon entropy in (1) is directly linked to the average number of binary questions (i.e., questions with a *yes* or *no* answer) one needs to ask to infer the state in which $X$ is. In our study, samples drawn from the population of the random variable $X$ are identified with values $Y_{r_i}$ and Shannon entropy can also be interpreted as a measure of the degree of heterogeneity of the system. In this sense, considering a support scale $r_i$, if the collected data (which are spatially distributed over the system) would cluster into one (or only a few) bin(s), one could interpret the system as homogeneous (or nearly homogeneous) at such a scale.

The information content shared by two random variables, i.e., $X_1$ and $X_2$, is termed bivariate mutual information and is defined as

$$I(X_1; X_2) = \sum_{i=1}^{N} \sum_{j=1}^{M} p_{i,j} \ln\left(\frac{p_{i,j}}{p_i p_j}\right) \tag{2}$$

where $N$ and $M$ represent the number of bins associated with $X_1$ and $X_2$, respectively; $p_i$ and $p_j$ are marginal probability mass functions associated with $X_1$ and $X_2$, respectively; and $p_{i,j}$ is the joint probability mass function of $X_1$ and $X_2$. The bivariate mutual information measures the average reduction in the uncertainty (as quantified through the Shannon entropy) about one random variable that one can obtain by knowledge on the other variable (Gong et al., 2013 and references therein). As such, the bivariate mutual information (*a*) vanishes for two independent variables and (*b*) coincides with the entropy of either of the two variables when one variable fully explains the other one, i.e., $H(X_2) = H(X_1) = I(X_1; X_2)$. In light of the latter observations, it is clear that the bivariate mutual information can be also interpreted as a measure of the degree of dependence between $X_1$ and $X_2$.

When considering three discrete random variables, it is possible to quantify the amount of information that two of these (termed as sources, i.e., $X_{S_1}$ and $X_{S_2}$) share with the third one (termed as target variable, i.e., $X_T$) upon evaluating the following multivariate mutual information

$$I(X_{S_1}, X_{S_2}; X_T) = \sum_{i=1}^{N} \sum_{j=1}^{M} \sum_{k=1}^{W} p_{i,j,k} \ln\left(\frac{p_{i,j,k}}{p_{i,j} p_k}\right) \tag{3}$$

Here, $N$, $M$, and $W$ represent the number of bins associated with $X_{S_1}$, $X_{S_2}$ and $X_T$, respectively; $p_k$ is the probability mass function of $X_T$; $p_{i,j}$ is the joint probability mass function of $X_{S_1}$ and $X_{S_2}$; and $p_{i,j,k}$ is the joint probability mass function of $X_{S_1}$, $X_{S_2}$, and $X_T$. Relying on the partial information decomposition or information partitioning (Williams and Beer, 2010;), the multivariate mutual information in (3) can be partitioned into unique, redundant, and synergetic contributions, i.e.,

$$I(X_{S_1}, X_{S_2}; X_T) = U(X_{S_1}; X_T) + U(X_{S_2}; X_T) + R(X_{S_1}, X_{S_2}; X_T) + S(X_{S_1}, X_{S_2}; X_T) \tag{4}$$

Here, $U(X_{S_1}; X_T)$ and $U(X_{S_2}; X_T)$ represent the amount of information that is uniquely provided to the target $X_T$ by $X_{S_1}$ and $X_{S_2}$, respectively (i.e., the information $U(X_{S_1}; X_T)$ cannot be provided to $X_T$ by knowledge on $X_{S_2}$, a corresponding observation holding for $U(X_{S_2}; X_T)$); the redundant contribution $R(X_{S_1}, X_{S_2}; X_T)$ is the information that both source variables provide to the target (i.e., it is the amount of information transferable to $X_T$ that is contained in both $X_{S_1}$ and $X_{S_2}$); and the synergetic contribution $S(X_{S_1}, X_{S_2}; X_T)$ is the information about $X_T$ that knowledge on $X_{S_1}$ and $X_{S_2}$ brings in a synergic way. Note that the latter contribution corresponds to the amount of information that (possibly) emerges by simultaneous knowledge of the two sources and through an analysis of their joint relationship with $X_T$, i.e., it would not appear by knowing both $X_{S_1}$ and $X_{S_2}$ while analyzing their individual relationship with $X_T$ separately. All components in (4) are positive (Williams and Beer, 2010). Figure 2 provides a graphical depiction in terms of Venn diagrams of the above information components in a system characterized by two sources and a target variable.

217       The bivariate mutual information shared by the target and each source can be written as

$$I(X_{S_1}; X_T) = U(X_{S_1}; X_T) + R(X_{S_1}, X_{S_2}; X_T)$$
$$I(X_{S_2}; X_T) = U(X_{S_2}; X_T) + R(X_{S_1}, X_{S_2}; X_T)$$
(5)

Note that (5) reflects the nature of the information that is shared by the target and each of the sources,
when these are taken separately, i.e., no synergy can be detected here. We also remark that one should
expect the emergence of some redundancy of information when the two sources are correlated.

222       An additional element of relevance for the aim of our study is the interaction information

$$I(X_{S_1}; X_{S_2}; X_T) = I(X_{S_1}; X_T \mid X_{S_2}) - I(X_{S_1}; X_T) =$$
$$= I(X_{S_2}; X_T \mid X_{S_1}) - I(X_{S_2}; X_T)$$
(6)

Here, $I(X_{S_i}; X_T \mid X_{S_j})$ is the bivariate mutual information shared by source $X_{S_i}$ ($i = 1, 2$) and the
target, conditional to the knowledge of source $X_{S_j}$ ($j = 2, 1$). Note that $I(X_{S_i}; X_T \mid X_{S_j})$ can be
evaluated in a way similar to (2) upon relying on the conditional probability for $X_T$. Williams and
Beer (2011) show that
$\quad I(X_{S_1}; X_{S_2}; X_T) = S(X_{S_1}, X_{S_2}; X_T) - R(X_{S_1}, X_{S_2}; X_T)$             (7)
According to (7), the bivariate interaction information could be either positive, i.e., when synergetic
interactions prevail over redundant contribution, or negative, i.e., when the degree of redundancy
overcomes the synergetic effects.

232       Inspection of (4)-(7) reveals that an additional equation is required to evaluate all components
in (4). Various strategies have been proposed in this context (e.g., Williams and Beer, 2010; Harder
et al., 2013; Bertschinger et al., 2014; Griffith and Koch, 2014; Olbrich et al., 2015; Griffith and Ho,
2015). We rest here on the recent partitioning strategy formalized by Goodwell and Kumar (2017),
due to its capability of accounting for the (possible) dependences between sources when evaluating
the unique and redundant contributions. The rationale underpinning this strategy is that (*i*) each of the
two sources can provide a unique contribution of information to the target even as these are correlated,
and (*ii*) redundancy should be lowest in case of independent sources. The redundant contribution can
then be evaluated as (Goodwell and Kumar, 2017)
$\quad R(X_{S_1}, X_{S_2}; X_T) = R_{\min}(X_{S_1}, X_{S_2}; X_T) + I_s(R_{MMI}(X_{S_1}, X_{S_2}; X_T) - R_{\min}(X_{S_1}, X_{S_2}; X_T))$     (8a)
with

$$R_{\min}(X_{S_1}, X_{S_2}; X_T) = \max(0, -I(X_{S_1}; X_{S_2}; X_T));$$
$$R_{MMI}(X_{S_1}, X_{S_2}; X_T) = \min(I(X_{S_2}; X_T), I(X_{S_1}; X_T));$$
$$I_s = \frac{I(X_{S_1}; X_{S_2})}{\min(H(X_{S_1}), H(X_{S_2}))};$$

(8b)

Goodwell and Kumar (2017) termed (8) as a rescaled measure of redundancy whereas (*a*)
$R_{\min}(X_{S_1}, X_{S_2}; X_T)$ represents the lowest bound for redundancy, which is set on the basis of the
rationale that the minimum value of redundancy must at least be equal to $-I(X_{S_1}; X_{S_2}; X_T)$ in case
$I(X_{S_1}; X_{S_2}; X_T) < 0$ (thus also ensuring positiveness of the synergy; see (7)); (*b*) $R_{MMI}(X_{S_1}, X_{S_2}; X_T)$
is an upper bound, consistent with the rationale that all information from the weakest source is
redundant; and (*c*) $I_s$ accounts for the degree of dependence between the sources, i.e., $I_s = 0$ and
$R(X_{S_1}, X_{S_2}; X_T) = R_{\min}(X_{S_1}, X_{S_2}; X_T)$ for independent sources, while $I_s = 1$ and redundancy in (8)
attains its upper limit value, $R_{MMI}(X_{S_1}, X_{S_2}; X_T)$, in case of a *complete* dependency (i.e.,
$X_{S_1} = f(X_{S_2})$ or vice versa) between the sources. Once the redundancy has been evaluated, all of the
other components in (4) can be determined.
We emphasize that, despite some additional complexities, analyzing the partitioning of the
multivariate mutual information provides valuable insights on the way information is shared across
three variables, these being here permeability data associated with three diverse support scales. In
summary, addressing information partitioning enables us to (*i*) quantify and (*ii*) characterize the
nature of the information that two variables (sources) provide to a third one (target) as a *whole*, i.e.,
considering the entire triplet. Doing so overcomes the limitation of depicting the system as a simple
*sum of parts*, as based on solely inspecting the corresponding pairwise bivariate mutual information,
which allows quantifying just the amount of information that pairs of variables (i.e., the first source
and the target; and the second source and the target) share (without being able to define redundant or
unique contributions, see Eq. (9)). In the context of our work, this implies that information
partitioning enables us to characterize the nature of the information that permeability data collected
at two support scales provide to /share with permeability data taken at a third one.

## 3.2 Implementation Aspects

Evaluation of the quantities introduced in Section 3.1 is accomplished according to three main
steps. We employ the Kernel Density Estimator (KDE) routines in Matlab2018© to estimate the
continuous counterparts of the probability mass functions $p_i$, $p_j$, $p_{i,j}$, and $p_{i,j,k}$ and assess the
associated probability density functions, i.e., *pdf*s. This step enables us to smooth and regularize the
available finite datasets. We then discretize the ensuing *pdf*s to evaluate the associated probability
mass functions. Note that this two-step procedure allows us to obtain results that are more stable (with
respect to the number of bins employed) than those that one could obtain upon discretizing directly
the available finite datasets. As a final step, we evaluate the metrics detailed in Section 3 by treating
separately the multi-scale measurements on each face and then averaging the ensuing face-related
results for each of the two rock samples. The benefit of resting on this approach is especially critical
when considering the Topopah rock, whereas pooling the data of all faces as a unique sample hindered
the emergence of the bimodal behavior (i.e., the permeability values corresponding to the peaks of
the bimodal distributions are slightly different depending on the face considered and the joint
treatment of the data from all faces yielded a nearly unimodal distribution). We employ a binning
scheme corresponding to a uniform discretization of the range delimited by the lowest and largest
values detected considering all datasets associated with both rocks (i.e., we employ the same specific
binning for the Berea and the Topopah rock samples to assist quantitative comparison of the results).
We observe that within an IT approach the selection of a bin size is an a priori choice (see, e.g., Gong
et al., 2014; Loritz et al., 2018) the influence of which should be properly assessed (see Section 4 and
Supplementary Materia1). We inspect how the IT metrics described in Section 2 vary as a function
of (*i*) the number of bins (i.e., we consider a number of 50, 75, 100, and 125 bins for the discretization
of the range of data variability) and (*ii*) the size of the kernel bandwidth (which is varied within the
range 0.1 - 0.4) employed in the KDE routine (see Supplementary Material, Figures SM1-3, for
additional details). This analysis highlights a weak dependence of the values of the investigate IT
metrics on the number of bins and on the size of the bandwidth employed in the Kernel Density
Estimator (KDE) procedure, the overall patterns of these metrics remaining substantially unaffected.
This leads us to use 100 bins and a kernel bandwidth equal to 0.3. Note that we consistently employ
this binning for the evaluation of all metrics introduced in Section 2.
We remark that the bivariate and multivariate mutual information metrics are evaluated by
focusing on the joint probability mass function grounded on the multi-scale data collected at the same
location on the sampling grids.

298                                    **4. Results**

Figure 3 depicts the probability mass function $p(Y_{r_i})$ for $i = 1$ ( $r_1$ ; black symbols), 2 ( $r_2$ ; red
symbols), 3 ( $r_3$ ; blue symbols), and 4 ( $r_4$ ; green symbols) for the (a) Berea and (b) the Topopah rock
samples. For both rocks the $p(Y_{r_i})$ associated with only one face is depicted (similar patterns are
noted for all of the remaining faces). Figure 3c depicts the Shannon entropy $H(Y_{r_i})$ as a function of
the MSP support scale $r_i$ for the Berea (diamonds) and the Topopah (circles) samples. Figure 3d
depicts the bivariate mutual information between data collected at two distinct support scales. This
metric is normalized by the entropy of the data associated with the smaller support scale, i.e.,
$I^*(Y_{r_i};Y_{r_j}) = I(Y_{r_i};Y_{r_j})/H(Y_{r_i})$ with $j > i$, for $i = 1$ (blue diamonds) and 2 (green diamonds), results for
the Berea (diamonds) and the Topopah (circles) samples are reported.
Inspection of Figure 3a-b reveals that distributions related to increasing values of $r_i$ tend not to
encompass extreme values (in particular the low ones) of $Y$. This observation supports the fact that
increasing $r_i$ favors a homogenization of the permeability values and suggests that the response of
the MSP tends to be only weakly sensitive to the less permeable portions of the rock that are
encompassed within a given measurement scale. As a consequence, the $p(Y_{r_i})$ associated with
increasing $r_i$ are characterized by a reduced number of populated bins, this feature being in turn
reflected in the observed reduction of $H(Y_{r_i})$ with increasing $r_i$ (Figure 3c) for both rock samples.
This result can be interpreted as a signature (see also the discussion about (1) in Section 3.1) of the
effect of increasing $r_i$, which yields a decrease of (*i*) the uncertainty about the spatial distribution of
the values of $Y_{r_i}$ and (*ii*) the ability of capturing the degree of spatial heterogeneity of $Y$. Note that
Figure 3c suggests that the value of $H(Y_{r_i})$, given $r_i$, associated with the Topopah sample is always
higher than its counterpart associated with the Berea rock. This outcome is consistent with the higher
heterogeneity displayed by the former sample, where the spatial distribution of $Y_{r_i}$ is affected by an
increased level of uncertainty as compared to its Berea-based counterpart.
Otherwise, two distinct behaviors emerge with regard to the location of the peak(s) of the
distributions: (*i*) the location of the peak of the distributions is virtually insensitive to $r_i$ for the Berea;
while (*ii*) the two peaks of the bimodal distributions of the Topopah sample display a clear tendency
to migrate towards higher permeability values as $r_i$ increases. These observations are consistent with
the homogeneous nature of the Berea and the two-material (pumice and matrix being high and low
permeable, respectively) type of heterogeneity displayed by the Topopah sample. It is also in line
with the previously noted weak sensitivity of the MSP measurements to region of low permeability.
With reference to the Berea sample, if a measurement taken at a given location with a small $r_i$ is close

to the average value (i.e., $Y_{r_i}$ is close to zero in our setting), it is likely that the same behavior is observed also for larger $r_i$ due to the homogeneity of the sample. Otherwise, in the case of the Topopah sample there are more chances that increasing $r_i$ (hence involving larger volumes of the rock) yields a shift of the ensuing measurements toward higher values.

Inspection of Figure 3d reveals that, given a reference support scale $r_i$, the mutual information shared with measurements taken at larger support scales $r_j$ decreases with increasing $r_j$ for both rock samples. In other words, the representativeness for system characterization of the sets of data associated with increasingly coarse support scale diminishes, as compared to the data collected at the given reference scale. At the same time, we note that the way in which $I^*(Y_{r_i};Y_{r_j})$ decreases with $r_j$ is very similar for (*i*) the two analyzed reference support scales, i.e., $r_1$ and $r_2$, and (*ii*) for the two considered rock types. We interpret this result as a sign of (at least qualitative) consistency in the way information is shared between datasets of measurements associated with increasing size of $r_i$, despite the different geological nature of the two types of samples analyzed. Otherwise, Figure 3d indicates that the (normalized) mutual information $I^*(Y_{r_i};Y_{r_j})$ is always lower in the Topopah than in the Berea system. This result provides a quantification of the qualitative observation that there is an overall decrease of the representativeness of the datasets associated with increasing data support (with respect to data collected with smaller $r_i$) as the system heterogeneity becomes stronger.

Figure 4 depicts the results of the information partitioning procedure detailed in Section 2.3 considering the Berea sample and two triplets of datasets $(Y_{r_{i+1}},Y_{r_{i+2}};Y_{r_i})$, with $r_i =$ (a) $r_1$ and (b) $r_2$. Corresponding results for the Topopah sample are depicted in (c) for $r_i = r_1$ and (d) for $r_i = r_2$. For ease of comparison between the results, we normalize the unique, synergetic and redundant contributions in (4) by the multivariate mutual information of the corresponding triplet, e.g.,

$$U^*(Y_{r_{i+1}};Y_{r_i}) = U(Y_{r_{i+1}};Y_{r_i})/I(Y_{r_{i+1}},Y_{r_{i+2}};Y_{r_i}), \qquad U^*(Y_{r_{i+2}};Y_{r_i}) = U(Y_{r_{i+2}};Y_{r_i})/I(Y_{r_{i+1}},Y_{r_{i+2}};Y_{r_i});$$

$$R^*(Y_{r_{i+1}},Y_{r_{i+2}};Y_{r_i}) = R(Y_{r_{i+1}},Y_{r_{i+2}};Y_{r_i})/I(Y_{r_{i+1}},Y_{r_{i+2}};Y_{r_i}), \quad S^*(Y_{r_{i+1}},Y_{r_{i+2}};Y_{r_i}) = S(Y_{r_{i+1}},Y_{r_{i+2}};Y_{r_i})/I(Y_{r_{i+1}},Y_{r_{i+2}};Y_{r_i}).$$

Results in Figure 4a-b suggest that for the Berea sample: (*i*) most of the multivariate information is redundant, a finding that can be linked to the dependence detected between the sets of data associated with the two coarser support scales (see, e.g., Figure 3d); (*ii*) the synergetic information is practically zero for both triplets considered, i.e., the simultaneous knowledge of the system at two coarser scales does not provide any additional information; (*iii*) data associated with the middle (in the triplets) support scale provides a non-negligible unique information content, the latter being less pronounced for the data referring to the most coarse support (in the triples). These results (i.e., high redundancy and high/low uniqueness for the middle/largest support scale) suggest that, considering the depiction of the system rendered at the finest support scale, the information provided by the investigations at the coarsest support scale is mostly contained by the information provided by the data collected at the intermediate scale. This element suggests a nested nature of the information linked to data collected at progressively increasing scales with respect to the information contained in the data associated with the smallest support scale. This finding can be linked to the homogeneous nature of the Berea sample, whereas the characterization at diverse scales does not change dramatically (e.g., note the similarities in the spatial patterns of $Y_{r_i}$ in Figure 1 for the Berea sample as a function of $r_i$), thus

promoting (*a*) the redundancy of information associated with measurements at the intermediate and
lager scales and (*b*) the uniqueness of information revealed for the intermediate scale.
Otherwise, inspection of Figure 4c-d reveals that for the Topopah rock sample: (*i*) most of the
multivariate information coincides with the unique information associated with the intermediate
scale; (*ii*) the redundant and unique contribution associated with the largest scale are still non-
negligible, yet being substantially smaller than the uniqueness contribution provided by the
intermediate scale; (*iii*) there is practically no synergetic information. This set of results descends
from the moderate or marked discrepancies displayed by $Y_{r_i}$ data as $r_i$ increases by one or two sizes,
respectively (e.g., see the faces depicted in Figure 1 for the Topopah sample). In other words, relying
on a device such as the MSP to obtain permeability data enables sampling a volume of the rock
according to which the majority of the multivariate information in a triplet is associated with a
significant unique contribution of the intermediate scale, the information related to the largest scale
still being weakly unique and weakly redundant.

## 5. Discussion

We recall that the focus of the present study is the quantification of the information content and
information shared between pairs and triplets of datasets of air permeability observations associated
with diverse sizes of the measurement/support scale. We exemplify our analysis upon relying on data
collected across two different types of rocks, i.e., a Berea and a Topopah sample, that are
characterized by different degrees of heterogeneity.
These datasets (or part of these) have been considered in some prior studies. Tidwell and Wilson
(1999a, b) and Lowry and Tidwell (2005) assess the impact of the size of the support/measurement
scale on key summary one-point (i.e., mean and variance) and two-points (i.e., variogram) statistics
within the context of classical geostatistical methods and evaluate kriging-based estimates of the
underlying random fields. Siena et al. (2012) and Riva et al. (2013) analyze the scaling behavior of
the main statistics of the log permeability data and of their increments (i.e., sample structure functions
of various orders), with emphasis on the assessment of power-law scaling behavior. On these bases,
Riva et al. (2013) conclude that the data related to the Berea sample can be interpreted as observations
from a sub-Gaussian random field subordinated to truncated fractional Brownian motion or Gaussian
noise. All of these studies focus on (*a*) the geostatistical interpretation of the behavior displayed by
the probability density function (and key moments) of the data and their spatial increments and (*b*)
the analysis of the skill of selected models to interpret the observed behavior of the main statistical
descriptors evaluated upon considering separately data associated with diverse measurement/support
scale. Furthermore, Tidwell and Wilson (2002) analyzed the Berea and Topopah datasets (considering
separately data characterized by diverse support scales) to assess possible correspondences between
the permeability field and some attributes of the rock samples determined visually through digital
imaging and conclude that image analysis can assist delineation of spatial patterns of permeability.
We remark that in all of the studies mentioned above the datasets associated with a given
support (or measurement) scale are analyzed separately. Otherwise, we leverage on elements of IT,
which allow a unique opportunity to circumvent limitations of linear metrics (e.g., Pearson
correlation) and analyze the relationships (in terms of shared amount of information) between pairs
(i.e., bivariate mutual information) or triplets (i.e., multivariate mutual information) of variables. We
also note that, even as visual inspection of $p(Y_{r_i})$ associated with diverse sizes of the support scale $r_i$
(see Figure 3a and Figure 3b for the Berea and Topopah, respectively) can show that these probability
densities can be intuitively linked to the documented decrease of the corresponding Shannon entropies

with increasing $r_i$ (see Figure 3c and Section 4), it would be hard to readily infer from such a visual comparative inspection the behavior of the bivariate (see Figure 3d) and multivariate (see Figure 4) mutual information because these require (see Eq.s (2)-(8)) the evaluation of the joint probability mass functions.

Considering an operational context, including, e.g., groundwater resource management or (conventional/unconventional) oil recovery, we observe that it is common to have at our disposal permeability data associated with diverse support scales. These can be inferred from, e.g., large scale pumping tests, downhole impeller flowmeter measurements, core flood experiments at the laboratory scale, geophysical investigations, or particle-size curves (see e.g., Paillet, 1989; Oliver, 1990; Dykaar and Kitanidis, 1992; Harvey, 1992; Deutsch and Journel, 1994; Day-Lewis et al., 2000; Zhang and Winter, 2000; Attinger, 2003; Pavelic et al., 2006; Neuman et al., 2008; Riva et al., 2099; Barahona-Palomo et al., 2011; Quinn et al., 2012; Shapiro et al., 2015; Galvão et al., 2016; Menafoglio et al., 2016; Medici et al., 2017; Dausse et al., 2019, and reference therein). Assessing (*i*) the information content and (*ii*) the amount of information shared between permeability data associated with differing support scales (and/or diverse measuring devices/techniques) along the lines illustrated in the present study can be beneficial to obtain a quantitative appraisal of possible feedbacks among diverse approaches employed for aquifer/reservoir characterization. Results of such an analysis can potentially serve as a guidance for the screening of datasets which are most informative to provide a comprehensive description of the spatially heterogeneous distribution of permeability. While the methodology detailed in Section 3 is readily transferable to scenarios where multi-scale permeability are available, the appraisal of the general nature of some specific findings of the present study (e.g., decrease of the Shannon entropy as the support scale increases, regularity in the trends displayed by the normalized bivariate mutual information) still remains an open issue which will be the subject of future works.

## 6. Conclusions

We rely on elements of Information Theory to interpret multi-scale permeability data collected over blocks of Berea Sandstone and a Topopah Spring Tuff, representing a nearly homogeneous and a heterogeneous porous medium composed of a two-material mixture, respectively. The unique multi-scale nature of the data enables us to quantify the way information is shared across measurement scales, clearly identifying information losses and/or redundancies that can be associated with the joint use of permeability data collected at differing scales. Our study leads to the following major conclusions:

1. An increase in the characteristic length associated with the scale at which the laboratory scale (normalized) gas permeability data are collected corresponds to a quantifiable decrease in the Shannon entropy of the associated probability mass function. This result is consistent with the qualitative observation that the ability of capturing the degree of spatial heterogeneity of the system decreases as the data support scale increases.

2. The (normalized) bivariate mutual information shared between pairs of permeability datasets collected at (*i*) a fixed fine scale (taken as reference) and (*ii*) larger scales decreases in a mostly regular fashion independent from the size of the reference scale, once the bivariate mutual information is normalized by the Shannon entropy of the data taken at the reference scale. This result highlights a consistency in the way information associated with data at diverse scales is shared for the instrument and the porous systems here analyzed.

3. As the degree of heterogeneity of the system increases, we document a corresponding increase in the Shannon entropy (given a support scale) and a decrease in the values of the

normalized bivariate mutual information (given two support scales) between permeability data collected at the differing measurement scales.

4. Results of the information partitioning of the multivariate mutual information shared by permeability data collected at three increasing support scales for the Berea sandstone sample exhibit a marked level of redundancy and high/low uniqueness for the data collected at the intermediate/coarser scale in the triplets with respect to the data associated with the finest scale. This result can be linked to the fairly homogeneous nature of the sample, that is also reflected in the moderate variation of the observed (normalized) gas permeability values with increasing size of the support scale.

5. Information partitioning for the Topopah tuff sample indicates the occurrence of a still significant amount of unique information associated with the data collected at the intermediate scale, while the redundant portion and the unique contribution linked to the largest scale in a triplet are clearly diminished. This result descends from the heterogeneous structure of the Topopah porous system, where the recorded (normalized) gas permeabilities display moderate or marked discrepancies as $r_i$ increases by one or two sizes, respectively.

6. For both rock samples considered, the simultaneous knowledge of permeability data taken at the intermediate and coarser support scales in a triplet does not provide significant additional information with respect to that already contained in the data taken at the fine scale, i.e., the synergic contribution in the resulting datasets is virtually zero.

Given the nature of the approach we employ, the latter is potentially amenable to be transferred to analyze settings involving other kinds of datasets associated with diverse hydrogeological quantities (including, e.g., porosity or sorption/desorption parameters) or considering measurement/sampling devices of a diverse design. Future developments could also include exploring the possibility of embedding the approach within the workflow of optimal experimental design and/or data-worth analysis strategies.

## Data Availability

Data employed were graciously provided by Tidwell, V.C., and are available online (https://data.mendeley.com/datasets/ygcgv32nw5/1).

## Author contributions

The methodology was developed by AD, supervised by and discussed with AG and MR. All codes were developed by AD. The manuscript was drafted by AD. Structure, narrative and language of the manuscript were revised and significantly improved by AG and MR.

## Competing interests

The authors declare to have no competing interests.

## Acknowledgements

The authors would like to thank the EU and MIUR for funding, in the frame of the collaborative international Consortium (WE-NEED) financed under the ERA-NET WaterWorks2014 Cofunded Call. This ERA-NET is an integral part of the 2015 Joint Activities developed by the Water Challenges for a Changing World Joint Programme Initiative (Water JPI). Prof. A. Guadagnini acknowledges funding from Région Grand-Est and Strasbourg-Eurométropole through the 'Chair Gutenberg'.

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

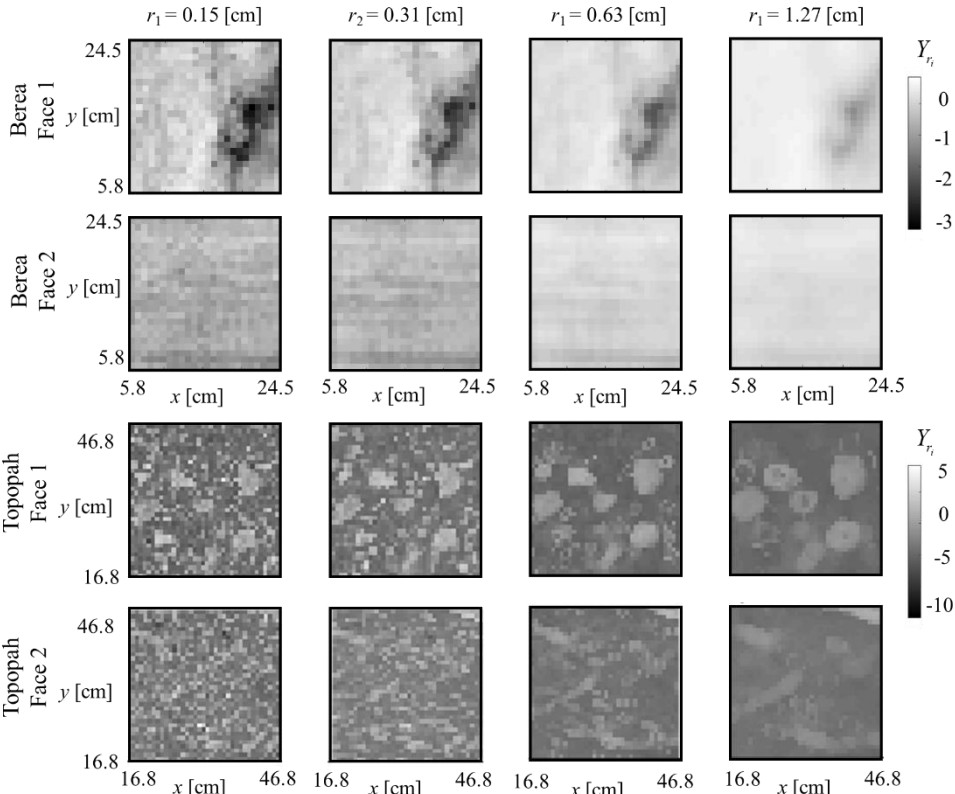

755

Figure 1. Examples of spatial distributions of the natural logarithm of normalized gas permeability, $Y_{r_i}$, for two faces of a cubic block of Berea Sandstone (first and second rows) and Topopah Spring Tuff (third and fourth rows) taken with four increasing support scales (columns, left to right).

759

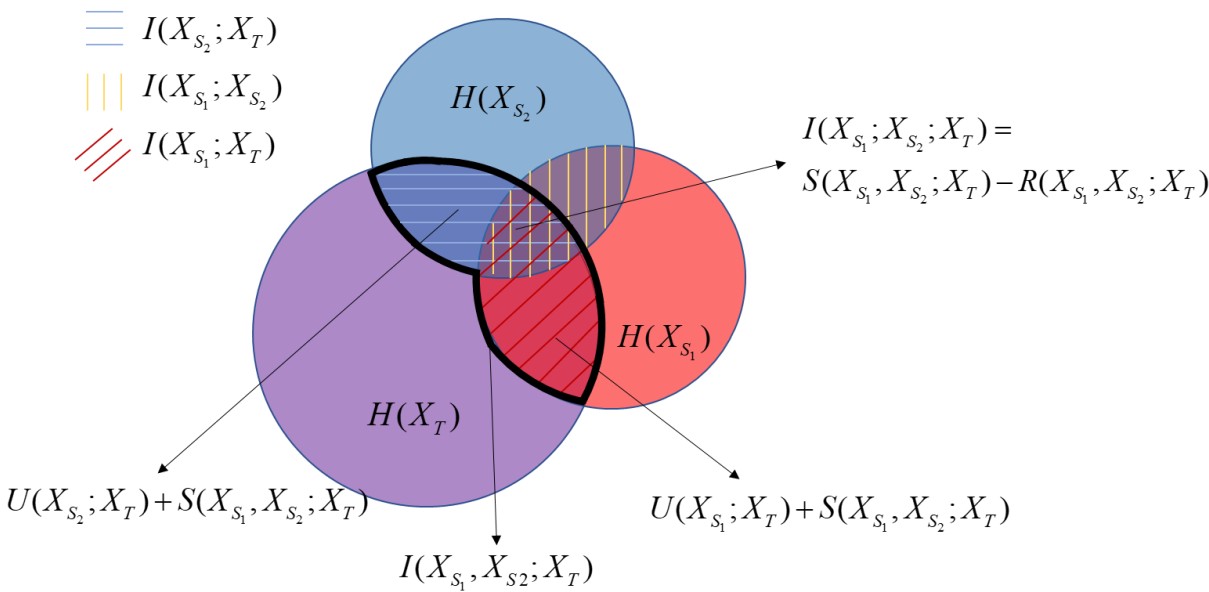

Figure 2. Venn diagram representation of the Information Theory concepts considering two sources, i.e., $X_{S_1}$ and $X_{S_2}$, and a target variable, $X_T$. Areas of the circles are proportional to Shannon entropy (i.e., $H(X_{S_1})$, $H(X_{S_2})$ and $H(X_T)$); overlaps of pairs of circles reflect bivariate mutual information (i.e., $I(X_{S_1};X_T)$, $I(X_{S_2};X_T)$, and $I(X_{S_1};X_{S_2})$); and the strength of the multivariate mutual information (i.e., $I(X_{S_1},X_{S_2};X_T)$) corresponds to the region delimited by the thick black curve. Unique (i.e., $U(X_{S_1};X_T)$ and $U(X_{S_2};X_T)$), synergetic (i.e., $S(X_{S_1},X_{S_2};X_T)$), and redundant (i.e., $R(X_{S_1},X_{S_2};X_T)$) components are also highlighted, as well as the interaction information (i.e., $I(X_{S_1};X_{S_2};X_T)$).

770

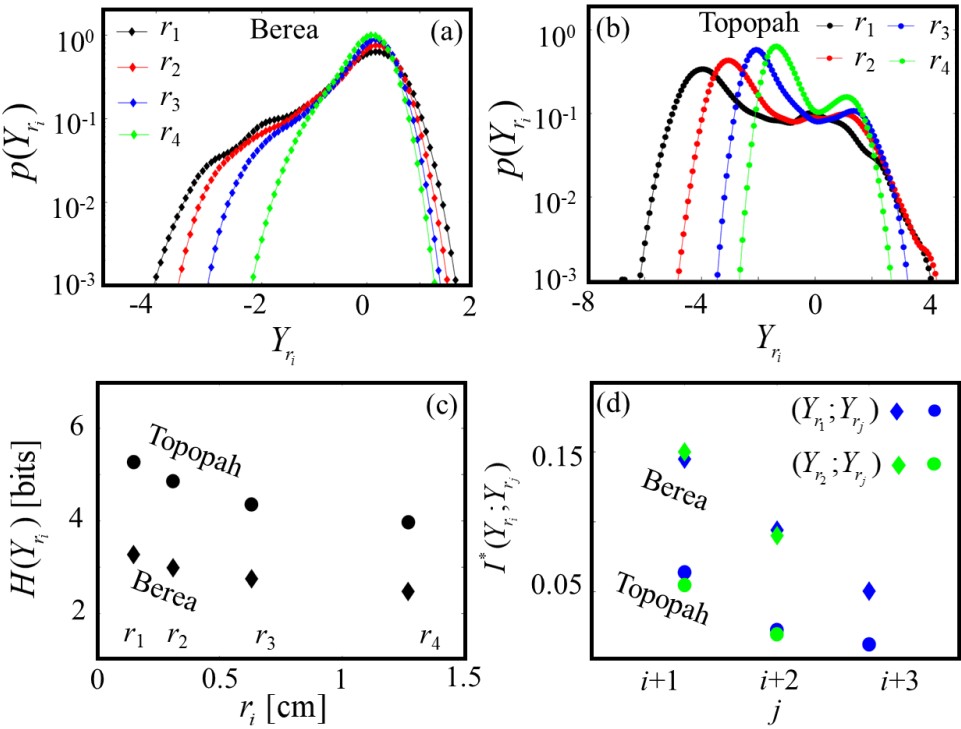

771

Figure 3. Probability mass function of the logarithm of normalized gas permeability, $p(Y_{r_i})$, for various support scale, $r_i$ ($i = 1$ (black), 2 (red), 3 (blue), 4 (green)) for (a) the Berea and (b) the Topopah samples; (c) Shannon entropy $H(Y_{r_i})$ versus $r_i$ for the Topopah (circles) and the Berea (diamonds) samples; (d) bivariate normalized mutual information $I(Y_{r_i};Y_{r_j})^* = I(Y_{r_i};Y_{r_j})/H(Y_{r_i})$ between data at a reference support scale, $Y_{r_i}$, and data at larger support scales, $Y_{r_j}$, for $i = 1$ (blue symbols), 2 (green simbols), considering the Berea (diamonds) and the Topopah (circles) rock samples.


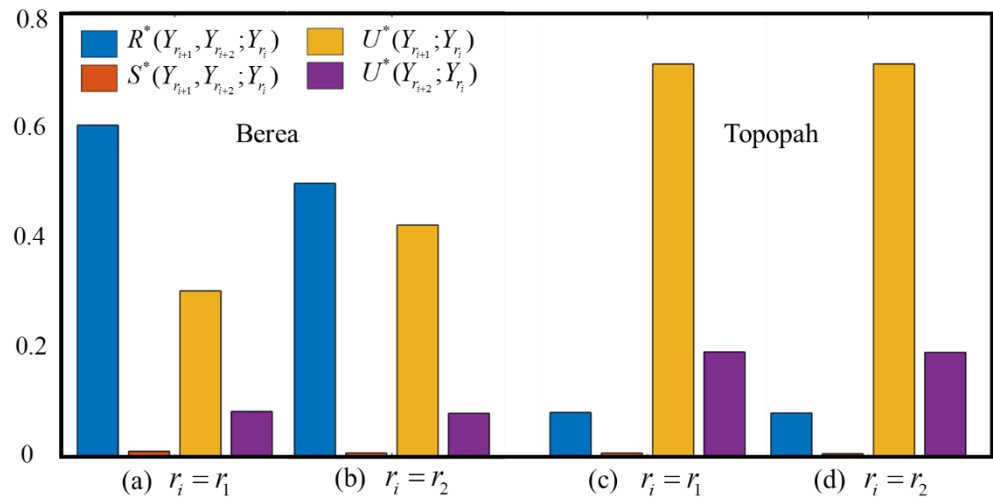


Figure 4. Information Partitioning of the multivariate mutual information, $I(Y_{r_{i+1}},Y_{r_{i+2}};Y_{r_i})$, considering
two triplets of data and $r_i =$ (a) $r_1$ and (b) $r_2$ for the Berea sample and $r_i =$ (c) $r_1$ and (d) $r_2$ for the
Topopah sample. For ease of comparison, we show the redundant, unique, and synergetic,
contributions normalized by $I(Y_{r_{i+1}},Y_{r_{i+2}};Y_{r_i})$.
