# Peer review of "1. Introduction"

_Hydrology and Earth System Sciences, 2019_

## Referee Comment (RC1) · Ralf Loritz (Referee) · 8 Jan 2020

**Review of "Interpretation of Multi-scale Permeability Data through an Information Theory Perspective" by Aronne Dell'Oca et al. (2019).**

**Summary and Recommendation**

In this manuscript (MS), the authors use a series of different methods taken from information theory to estimate and compare the information content of different permeability measurements of two geological settings.

Overall, the MS is structure and written in a clear manner. In addition, the general idea behind the study is interesting and of relevance for the potential readers of HESS. However, I have to admit that I was a bit surprised because I could not find a scientific discussion in the entire MS. While there a few interpretations of the results in section 4 there is not a single reference after page 7 neither a discussion. This left me with a lot of open questions like:

> How does your results link to the work of Tidwell and other who used for instance geo-statistics on the same data set.
>
> Which findings are new and could have not been drawn if you have used of more classical statistical approaches instead of Information theory?
>
> What are the merits of using Information Theory if most of your conclusions can be drawn by looking at the pdfs in figure 3 a.

Given the nicely written introduction and the overall interesting topic of the MS I am however very positive that the authors are able to re-work this MS in a way that it can be published in HESS. For this, I believe, however, that a substantial amount of work needs to be put in this MS before it can be published.

**Major comments:**

No scientific discussion and comparison with other research.

**Technical comments:**

**Section 3.2 Implementation Aspect Line 247-269:** Here, the authors chose a couple of crucial parameters, which are in my opinion not all well justified. For instance, they use a kernel density estimator to estimate their pdfs from their datasets. However, they give not much details how they chose their related parameters, neither how changing them influences the results nor why they do this besides stating that: "*This step enables us to smooth and regularize the available finite datasets*".

How do you know that you do not smoothed out information that is of relevance?

Furthermore, why do you chose 100 bins. Is this choice based on, for instance, the measurement uncertainties (physics; e.g. *Loritz et al. 2018*) or on a statistical analysis (statistics; e.g. *Gong et al. 2013*)? How do your results change if you only pick 50 bins? Remember that the bin width is pretty much your a-priori assumption of similarity so you need to be careful here.

**Line 83:** Well, again you need to choose your bin width, which is a strong a-priori assumption.

**Line 106:** Information is always about something. Please be more specific here.

**Line 155:** The formula is correct but rather uncommon in this form.

**Line 162:** The nature of the Shannon entropy does not change if you use *nats,* however, I would argue that the interpretation is much more straightforward if you use the binary logarithm calculate it. This is the case because it is then directly linked to the average number of binary questions one needs to ask to infer in which state X is as well as it is then directly linked to the maximum compressibility of your dataset. A perfect lossless compression is thereby a perfect upscaling.

**Line 244 – 246:** Why? Could you explain that in your specific context.

**References:**

Gong, W., Yang, D., Gupta, H. V. and Nearing, G.: Estimating information entropy for hydrological data: One-dimensional case, Water Resour. Res., 50(6), 5003–5018, doi:10.1002/2014WR015874, 2014.

Loritz, R., Gupta, H., Jackisch, C., Westhoff, M., Kleidon, A., Ehret, U. and Zehe, E.: On the dynamic nature of hydrological similarity, Hydrol. Earth Syst. Sci., 22(7), 3663–3684, doi:10.5194/hess-22-3663-2018, 2018.

---

## Referee Comment (RC2) · Anonymous Referee #2 · 18 Feb 2020

The authors use well-known information-theoretic quantities to quantify information content and information transfer among permeability datasets collected at different scales. The explanation of the quantities is thorough, but it is not clear to which extent the presented results are affected by the choice of the settings for the methodology (binning, bandwidth...) or how the information extracted from the datasets can be used in practice. I advise the authors to carefully review the manuscript, expanding the investigation to the analysis of the impact of "setting parameters" and presenting some ideas on the practicality of the analysis.

Specific comments - please investigate the role of binning with respect to the presented results - how do you choose the bandwidth? Does it have an influence on the results? - does the fact that permeability is by its nature a process-dependent (or model-dependent) quantity affect the applicability of the procedure? - could you please discuss: - how often multi-scale permeability measurements are available - how the presented results are transferable - how the presented results can be used in practical applications - lines 85-86: please expand the literature review to include several works on the use of information-theory quantities for porous material characterization -lines 145-147: please clarify meaning and implications

Technical comments A few typos: line 284, line 254

---

## Author Comment (AC1) · 15 Mar 2020

Interactive comment by Ralf Loritz on "Interpretation of Multi-scale Permeability Data through an Information Theory Perspective" by Aronne Dell'Oca, Monica Riva and Alberto Guadagnini (https://doi.org/10.5194/hess-2019-628).

Dear Editor:

We appreciate the efforts that you and Dr. Ralf Loritz have invested in our manuscript. We here detail the actions we envision to address Dr. Ralf Loritz's comments and inputs. Please, find below an item by item list whereas our envisioned actions are in-

[Figure]

dicated in plain font, to distinguish them from the Reviewer's comments (in italic). Our revised manuscript will be uploaded at a later stage, following closure of the discussions phase.

Summary and Recommendation

In this manuscript (MS), the authors use a series of different methods taken from information theory to estimate and compare the information content of different permeability measurements of two geological settings. Overall, the MS is structure and written in a clear manner. In addition, the general idea behind the study is interesting and of relevance for the potential readers of HESS. We thank Dr. Ralf Loritz for his appreciation of our work. However, I have to admit that I was a bit surprised because I could not find a scientific discussion in the entire MS. While there a few interpretations of the results in section 4 there is not a single reference after page 7 neither a discussion.

We thank Dr. Loritz for his comment. We have added the following Discussion section in the revised manuscript.

This left me with a lot of open questions like: How does your results link to the work of Tidwell and other who used for instance geo-statistics on the same data set. Which findings are new and could have not been drawn if you have used of more classical statistical approaches instead of Information theory? We thank Dr. Ralf Loritz for his comment. We will enhance the focus on the nature of the results of our analysis and the one performed by Tidwell and Wilson (1999a, b), Lowry and Tidwell (2005), Riva et al. (2013), Siena et al. (2012) and Tidwell and Wilson (2002).

Our discussion section includes the following text: 'We recall that the focus of the present study is the quantification of the information content and information shared between pairs and triplets of datasets of air permeability observations associated with diverse sizes of the measurement/support scale. We exemplify our analysis considering data collected across two different types of rocks, i.e., a Berea and a Topopah sample, that are characterized by different degrees of heterogeneity. These datasets

(or part of these) have been considered in some prior studies. Tidwell and Wilson (1999a, b) and Lowry and Tidwell (2005) assess the impact of the size of the support/measurement scale on key summary one-point (i.e., mean and variance) and two-points (i.e., variogram) statistics within the context of classical geostatistical methods and evaluated kriging-based estimates of the underlying random fields. Siena et al. (2012) and Riva et al. (2013) analyze the scaling behavior of the main statistics of the log permeability data and of their increments (i.e., sample structure functions of diverse orders), with emphasis on the assessment of power-law scaling behavior. On these bases, Riva et al. (2013) conclude that the data related to the Berea sample can be interpreted as observations from a sub-Gaussian random field subordinated to truncated fractional Brownian motion or Gaussian noise. All of these studies focus on (a) the geostatistical interpretation of the behavior displayed by the probability density function (and key moments) of the data and their spatial increments and (b) the analysis of the skill of selected models to interpret the observed behavior of the main statistical descriptors evaluated upon considering separately data associated with diverse measurement/support. Furthermore, Tidwell and Wilson (2002) analyzed the Berea and Topopah datasets (considering separately data characterized by diverse support scales) to assess possible correspondences between the permeability field and some attributes of the rock samples determined visually through digital imaging and conclude that image analysis can assist delineation of spatial permeability patterns. We remark that in all of the studies mentioned above the datasets associated with a given support (or measurement) scale are analyzed separately. Otherwise, we leverage on elements of IT which allow a unique opportunity to circumvent limitations of linear metrics (e.g., Pearson correlation) and analyze the relationships (in terms of shared amount of information) between pairs (i.e., bivariate mutual information) or triplets (i.e., multivariate mutual information) of variables.'

What are the merits of using Information Theory if most of your conclusions can be drawn by looking at the pdfs in figure 3a.

We thank Dr. Ralf Loritz for his comment and apologize for our lack of clarity. We refer to these pdfs to show consistency with our results and findings related to the analysis of the information shared between pairs and triplets of permeability datasets. These cannot otherwise be drawn from the mere inspection of the pdfs in Fig. 3a. Our revised manuscript includes the following text: 'We also note that, even as visual inspection of associated with diverse sizes of the support scale (see Figure 3a and Figure 3b for the Berea and Topopah, respectively) can show that these probability densities can be intuitively linked to the documented decrease of the corresponding Shannon entropies with increasing (see Figure 3c and Section 4), it would be hard to readily infer from such a visual comparative inspection the behavior of the bivariate (see Figure 3d) and multivariate (see Figure 4) mutual information because these require (see Eq.s (2)-(8)) the evaluation of the joint probability mass functions.'

Given the nicely written introduction and the overall interesting topic of the MS I am however very positive that the authors are able to re-work this MS in a way that it can be published in HESS. For this, I believe, however, that a substantial amount of work needs to be put in this MS before it can be published.

We thank Dr. Loritz for his appreciation of our work.

Major comment

No scientific discussion and comparison with other research.

We plan to add a Discussion section in the revised manuscript, where we will detail comparison with prior research relying on these data according to our answer above.

Technical comments

Section 3.2 Implementation Aspect Line 247-269: Here, the authors chose a couple of crucial parameters, which are in my opinion not all well justified. For instance, they use a kernel density estimator to estimate their pdfs from their datasets. However, they give not much details how they chose their related parameters, neither how changing them

influences the results nor why they do this besides stating that: "This step enables us to smooth and regularize the available finite datasets". How do you know that you do not smoothed out information that is of relevance? Furthermore, why do you chose 100 bins. Is this choice based on, for instance, the measurement uncertainties (physics; e.g. Loritz et al. 2018) or on a statistical analysis (statistics; e.g. Gong et al. 2013)? How do your results change if you only pick 50 bins? Remember that the bin width is pretty much your a-priori assumption of similarity so you need to be careful here.

We thank Dr. Loritz for his comment. The revised manuscript will include additional details on the selection of the parameter of the kernel density estimator (KDE) and on the number of bins. In summary, we consider various values of the KDE parameter (i.e., width of the kernel) and bin number to ensure stability of results. We note that evaluating the probability mass function directly from data led to unstable results, mainly due to the limited extent of the available datasets. To circumvent this drawback, we opt for a KDE approach to (a) infer the required pdfs and then (b) evaluate the discretized probability mass function. We clarify this aspect in the revised manuscript by adding the following text: 'We inspect how the IT metrics described in Section 2 vary as a function of (i) the number of bins (i.e., we consider a number of 50, 75, 100, and 125 bins for the discretization of the range of data variability) and (ii) the size of the kernel bandwidth (which is varied within the range 0.1 - 0.4) employed in the KDE routine (see Supplementary Material SM1-3 for additional details). This analysis highlights a weak dependence of the values of the investigate IT metrics on the number of bins and on the size of the bandwidth employed in the Kernel Density Estimator (KDE) procedure. However, the overall patterns of these metrics remain substantially unaffected. This leads us to use 100 bins and a kernel bandwidth equal to 0.3. Note that we consistently employ this binning for the evaluation of all metrics introduced in Section 2. Additional details on the impact of the number of bins and on the size of the kernel bandwidth will be provided as Supplementary Material.

Line 83: Well, again you need to choose your bin width, which is a strong a-priori

assumption.

We will point out this aspect in the revised manuscript where we write: We observe that within an IT approach the selection of a bin size is an a priori choice (see, e.g., Gong et al., 2014; Loritz et al., 2018) the influence of which should be properly assessed (see Section 4 and Supplementary Materia1).

Line 106: Information is always about something. Please be more specific here.

Our revised text now reads: 'we leverage on such a unique and truly multiscale datasets to address research questions such as "How much information about the natural logarithm of (normalized) gas permeabilities is lost as the support scale increases?".'

Line 155: The formula is correct but rather uncommon in this form.

We adopt this format to emphasize the concept of information, i.e., , as its interpretation as a degree of surprise. We will abide by the Editor on this element.

Line 162: The nature of the Shannon entropy does not change if you use nats, however, I would argue that the interpretation is much more straightforward if you use the binary logarithm calculate it. This is the case because it is then directly linked to the average number of binary questions one needs to ask to infer in which state X is as well as it is then directly linked to the maximum compressibility of your dataset. A perfect lossless compression is thereby a perfect upscaling.

We agree with Dr. Loritz and his interpretation of Shannon entropy when a base-two logarithm is employed. Otherwise, this will just affect the presentation of the results in Fig. 3a, whereas all of our remaining results are based on normalized quantities and the general conclusions and observations remain unaffected. We will modify the manuscript by employing base two logarithm. Our revised text now reads: 'We employ base two logarithms in (1), thus leading to bits as unit of measure for entropy and for the IT metrics we describe in the following. While other choices (relying, e.g., on the natural logarithm) are admissible, the nature and meaning of the metrics we illustrate

remain unaffected.' Furthermore, we will add the interpretation suggested by Dr. Loritz and write: 'Moreover, one can note that Shannon entropy in (1) is directly linked to the average number of binary questions (i.e., questions with a yes or no answer) one needs to ask to infer the state in which X is'.

Line 244 – 246: Why? Could you explain that in your specific context.

We thank Dr. Ralf Loritz for his comment. We will further elaborate on this concept in the revised manuscript, which now reads: 'In summary, addressing information partitioning enables us to (i) quantify and (ii) characterize the nature of the information that two variables (sources) provide to a third one (target) as a whole, i.e., considering the entire triplet. Doing so overcomes the limitation of depicting the system as a simple sum of parts, as based on solely inspecting the corresponding pairwise bivariate mutual information, which allows quantifying just the amount of information that pairs of variables (i.e., the first source and the target; and the second source and the target) share (without being able to define redundant or unique contributions, see Eq. (9)). In the context of our work, this implies that information partitioning enables us to characterize the nature of the information that permeability data collected at two support scales provide to /share with permeability data taken at a third one.'

References

Lowry, T. S., and Tidwell, V. C.: Investigation of permeability upscaling experiments using deterministic modeling and monte carlo analysis, World Water and Environmental Resources Congress 2005, May 15-19, Anchorage, Alaska, United States. https://doi.org/10.1061/40792(173)372, 2005.

Tidwell, V. C., and Wilson, J. L.: Permeability upscaling measured on a block of Berea Sandstone: Results and interpretation, Math. Geol., 31(7), 749-769, https://doi.org/10.1023/A:1007568632217, 541 1999a.

Tidwell, V. C., and Wilson, J. L.: Upscaling experiments conducted on a block of vol-

canic tuff: Results for a bimodal permeability distribution, Water Resour. Res., 35(11), 3375-3387, https://doi.org/10.1029/1999WR900161, 1999b.

Tidwell, V. C., and Wilson, J. L.: Visual attributes of a rock and their relationship to permeability: A comparison of digital image and minipermeameter data, Water Resour. Res., 38(11), 1261. doi:10.1029/2001WR000932, 2002.

Riva, M., Neuman, S. P., Guadagnini, A., and Siena, S.: Anisotropic scaling of Berea sandstone log air permeability statistics, Vadose Zone J., 12, 1-15. doi:10.2136/vzj2012.0153, 2013.

Siena, M., Guadagnini, A., Riva, M., and Neuman, S. P.: Extended power-law scaling of air permeabilities measured on a block of tuff, Hydrol. Earth Syst. Sci., 16, 29-42, 2012. doi:10.5194/hess-16-29-2012.

Please also note the supplement to this comment:
https://www.hydrol-earth-syst-sci-discuss.net/hess-2019-628/hess-2019-628-AC1-supplement.pdf
* * *

---

## Author Comment (AC2) · 15 Mar 2020

Interactive comment by anonymous reviewer on "Interpretation of Multi-scale Permeability Data through an Information Theory Perspective" by Aronne Dell'Oca, Monica Riva and Alberto Guadagnini (https://doi.org/10.5194/hess-2019-628).

Dear Editor:

We appreciate the efforts that you and the anonymous Reviewer have invested in our manuscript. We here detail the actions we envision to address the Reviewer's comments and inputs. Please, find below an item by item list where our envisioned actions

are indicated in plain font, to distinguish them from the Reviewer's comments (in italic). Our revised manuscript will be uploaded following closure of the discussions phase.

Summary and Recommendation

The authors use well-known information-theoretic quantities to quantify information content and information transfer among permeability datasets collected at different scales. The explanation of the quantities is thorough, but it is not clear to which extent the presented results are affected by the choice of the settings for the methodology (binning, bandwidth, . . .) or how the information extracted from the datasets can be used in practice. I advise the authors to carefully review the manuscript, expanding the investigation to the analysis of the impact of "setting parameters" and presenting some ideas on the practicality of the analysis.

We thank the Reviewer for his/her efforts and time. We will provide additional details on the impact of the number of bins and the size of the bandwidth of the kernel (i) in the manuscript and (ii) as supplementary material (in details). We will add a discussion section in the revised manuscript where we clarify the potential use and transferability of the current analysis in the context of practical applications.

Specific comments

Please investigate the role of binning with respect to the presented results - how do you choose the bandwidth? Does it have an influence on the results?

We thank the Reviewer for pointing this out. We will provide additional details on the assessment of the impact of the number of bins and of the size of the kernel bandwidth on the presented results. Our revised text now reads: "We inspect how the IT metrics described in Section 2 vary as a function of (i) the number of bins (i.e., we consider a number of 50, 75, 100, and 125 bins for the discretization of the range of data variability) and (ii) the size of the kernel bandwidth (which is varied within the range 0.1 - 0.4) employed in the KDE routine (see Supplementary Material SM1-3 for additional
details). This analysis highlights a weak dependence of the values of the investigate IT metrics on the number of bins and on the size of the bandwidth employed in the Kernel Density Estimator (KDE) procedure. However, the overall patterns of these metrics remain substantially unaffected. This leads us to use 100 bins and a kernel bandwidth equal to 0.3. Note that we consistently employ this binning for the evaluation of all metrics introduced in Section 2.". We will also include all details about these issues as supplementary material.

Does the fact that permeability is by its nature a process-dependent (or model-dependent) quantity affect the applicability of the procedure?

We do not see why the nature of permeability, including its scale dependence as an effective parameter associated with the flow equation, should hamper the applicability of the procedure. This is also in line with the consolidated use of standard geostatistical approaches for the stochastic characterization of heterogeneity of aquifer systems.

Could you please discuss: - how often multi-scale permeability measurements are available - how the presented results are transferable - how the presented results can be used in practical applications

We thank the Reviewer his/her comment. We will address these aspects by adding relevant references. Our revised text now reads (Section 5): "Considering an operational context, including, e.g., groundwater resource management or (conventional/unconventional) oil recovery, we observe that it is common to have at our disposal permeability data associated with diverse support scales. These can be inferred from, e.g., large scale pumping tests, downhole impeller flowmeter measurements, core flood experiments at the laboratory scale, geophysical investigations, or particle-size curves (see e.g., Paillet, 1989; Day-Lewis et al., 2000; Zhang and Winter, 2000; Pavelic et al., 2006; Neuman et al., 2008; Riva et al., 2099; Barahona-Palomo et al., 2011; Quinn et al., 2012; Shapiro et al., 2015; Galvão et al., 2016; Menafoglio et al., 2016; Medici et al., 2017; Dausse et al., 2019, and reference therein). Assessing (i) the information

content and (ii) the amount of information shared between permeability data associated with differing support scales (and/or diverse measuring devices/techniques) along the lines illustrated in the present study can be beneficial to obtain a quantitative appraisal of possible feedbacks among diverse approaches employed for aquifer/reservoir characterization. Results of such an analysis can potentially serve as a guidance for the screening of datasets which are most informative to provide a comprehensive description of the spatially heterogeneous distribution of permeability. While the methodology detailed in Section 3 is readily transferable to scenarios where multi-scale permeability are available, the appraisal of the general nature of some specific findings of the present study (e.g., decrease of the Shannon entropy as the support scale increases, regularity in the trends displayed by the normalized bivariate mutual information) still remains an open issue."

Lines 85-86: please expand the literature review to include several works on the use of information-theory quantities for porous material characterization.

We thank the Reviewer his/her comment. Our revised text now reads (Section 1): "To the best of our knowledge, as compared to surface hydrology systems only a limited set of works consider relying on IT concepts to analyze scenarios related to processes taking place in subsurface porous media. Nevertheless, we note a great variety in the topics covered in these works, reflecting the broad applicability of IT concepts. These studies include, e.g., the works of Woodbury and Ulrych (1993, 1996, 2000) who apply the principle of minimum relative entropy to tackle uncertainty propagation and inverse modeling in a groundwater system. The principle of maximum entropy is employed by Gotovac et al. (2010) to characterize the probability distribution function of travel time of a solute migrating within a heterogeneous porous formation. Within the same context, Kitanidis (1994) leverage on the definition of entropy and introduced the concept of dilution index to quantify the dilution state of a solute cloud migrating within an aquifer. Mishra et al. (2009) and Zeng et al. (2012) evaluate the mutual information shared between pairs of (uncertain) model input(s) and output(s) of interest, and
view this metric as a measure of global sensitivity. Nowak and Guthke (2016) focus on sorption of metals onto soil and the identification of an optimal experimental design procedure in the presence of multiple models to describe sorption. Boso and Tartakovsky (2018) illustrate an IT approach to upscale/downscale equations of flow in synthetic settings mimicking heterogeneous porous media. Relaying on IT metrics, Butera et al. (2018) assess the relevance of non-linear effects for the characterization of the spatial dependence of flow and solute transport related observables. Bianchi and Pedretti (2017, 2018) developed novel concepts, mutuated by IT, for the characterization of heterogeneity within a porous system and its links to salient solute transport features. Wellman and Regenaur-Lieb (2012) and Wellman (2013) leverage on IT concepts to quantify uncertainty, and its reduction, about the spatial arrangement of geological units of a subsurface formation. Recently, Mälicke et al. (2019) combine geostatistic and IT to analyze soil moisture data (representative of a given measurement scale) to assess the persistence over time of the spatial organization the soil moisture, under diverse hydrological regimes".

Lines145-147: please clarify meaning and implications

We thank the Reviewer his/her comment. We have further clarified our choice. Our revised text now reads: "While corresponding definitions are available also for continuous variables (i.e., summation(s) and probability mass function(s) are replaced by integral(s) and probability density function(s), respectively), these are characterized by a less intuitive and immediate interpretation (e.g., Entropy could be negative, infinite or could not be evaluated in case of probability density function(s) involving a Dirac's delta since its logarithm is not defined; see e.g., Cover and Thomas, 2006; Kaiser and Schreiber, 2002). Moreover, in case no analytical expressions are available for the demanded probability density functions of the analyzed continuous variables, a quantization of the latter is necessary in order to estimate the IT metrics associated with the continuous variables through their quantized counterparts (see Cover and Thomas, 2006). In general, the quality of these estimates increases (in different manners depending on the specific metric) with the level of quantization of the continuous variables (see e.g., Kaiser and Schreiber, 2002)."

Technical comments A few typos: line 284, line 254. We thank the Reviewer his/her comment. We will duly correct the typos in the revised manuscript.

References

[revised manuscript text omitted]

Please also note the supplement to this comment:
https://www.hydrol-earth-syst-sci-discuss.net/hess-2019-628/hess-2019-628-AC2-supplement.pdf